# Collisionless relaxation of a disequilibrated current sheet and implications for bifurcated structures

Young Dae Yoon [1✉], Gunsu S. Yun [2✉], Deirdre E. Wendel [3] & James L. Burch [4]

Current sheets are ubiquitous plasma structures that play the crucial role of being energy sources for various magnetic phenomena. Although a plethora of current sheet equilibrium solutions have been found, the collisionless process through which a disequilibrated current sheet relaxes or equilibrates remains largely unknown. Here we show, through analyses of phase-space distributions of single-particle orbit classes and particle-in-cell simulations, that collisionless transitions among the orbit classes are responsible for this process. Bifurcated current sheets, which are readily observed in geospace but whose origins remain controversial, are shown to naturally arise from the equilibration process and thus are likely to be the underlying structures in various phenomena; comparisons of spacecraft observations to particle-in-cell simulations support this fact. The bearing of this result on previous explanations of bifurcated structures is also discussed.

[1] Pohang Accelerator Laboratory, POSTECH, Pohang, Republic of Korea. [2] Department of Physics and Division of Advanced Nuclear Engineering, POSTECH, Pohang, Republic of Korea. [3] NASA Goddard Space Flight Center, Greenbelt, MD, USA. [4] Southwest Research Institute, San Antonio, TX, USA.
✉email: yyoon11@postech.ac.kr; gunsu@postech.ac.kr

Current sheets are structures generated by opposing magnetic fields and are ubiquitous in magnetized plasmas such as solar flares[1], the solar wind[2], the heliosphere[3,4], and planetary magnetospheres[5,6]. They are crucial localities where magnetic free energy can be converted to other forms of energy. Some important examples of this conversion include magnetic reconnection[7,8], the drift kink instability[9], and the tearing/plasmoid instability[10,11]. Sheet structures per se are also important in understanding, e.g., magnetosphere–ionosphere coupling[12] and the solar cycle[3]. Current sheets have thus been subject to extensive research, and a plethora of equilibrium solutions have been found both analytically[13–20] and numerically[21–24].

However, there remains an important outstanding question regarding current sheet equilibria. Although various equilibrium solutions have been found, the collisionless process through which a disequilibrated current sheet equilibrates remains largely unknown. The comprehension of this process is critical because plasma systems in general do not start from equilibria, and so they are predestined to relax toward a minimum energy state. Such knowledge therefore elucidates how a given system wants to evolve in time, even if it does not eventually equilibrate. In addition, the equilibria that have been found are specific solutions; a comprehensive understanding of the equilibration or relaxation process is therefore necessary in order to place current sheets in a general context.

A commonly observed form of current sheets is a bifurcated current sheet, which has two current density peaks on either side of the symmetry plane. These were first observed in the Earth's magnetotail by Cluster spacecraft measurements[25,26] and were initially deemed atypical. Later analyses, however, showed that bifurcated current sheets are actually extremely common, and that they were detected ~25% of the time Cluster was in the magnetotail current sheet[27,28]. These recurrent observations were puzzling because a current sheet had originally been thought to involve a single current peak providing the pinching force that opposes the diverging force due to the plasma thermal pressure; the existence of bifurcated structures thus implied that such naively macroscopic pictures are insufficient and more detailed analyses are warranted. Since then, various explanations have been put forth, including flapping motion[29], magnetic reconnection[30,31], temperature anisotropy[32,33], Speiser motion[34], and non-adiabatic scattering of particles in a strongly curved magnetic field[20]. However, there is no consensus on the origin of bifurcated current sheets, which largely remains a mystery despite being readily observed even to this day[35–37].

In this paper, the collisionless relaxation process of an initially disequilibrated current sheet is studied. The process is shown in three steps. First, particle orbits in a magnetic field reversal are comprehensively categorized into four orbit classes. Second, the phase-space distribution of each orbit class and the role each class plays with respect to current sheet density, temperature, and strength are examined. Finally, with the aid of particle-in-cell simulations, it is shown that transitions among the orbit classes are responsible for collisionless current sheet relaxation. The final equilibrium is most naturally understood in terms of the relative population of the phase-space distributions of the four orbit classes, instead of closed-form functions such as a Maxwellian. The bearing of this process on the origin of bifurcated current sheets is then discussed. Two of the orbit classes necessarily exhibit spatially bifurcated structures, and so such structures naturally arise as a current sheet evolves towards equilibrium via orbit class transitions. An exemplary equilibrium from particle-in-cell simulations is compared with Magnetospheric Multiscale (MMS) measurements of an electron-scale current sheet, and their profiles are shown to agree well. The relevance of the relaxation process to previous explanations of bifurcated current sheets is also discussed.

## Results

**Particle orbit classes.** Let us first examine single-particle dynamics in the renowned Harris current sheet[13], which is chosen as the system of scrutiny in the present study. It is described by the following magnetic field profile and distribution function $f_\sigma$ for species $\sigma$ (i for ions and e for electrons):

$$\mathbf{B}(x) = \hat{y} B_0 \tanh \frac{x}{\lambda}, \tag{1}$$

$$f_\sigma(x, \mathbf{v}, t) = \left(\frac{1}{2\pi v_{T\sigma}^2}\right)^{3/2} \frac{n_0}{\cosh^2(x/\lambda)}$$
$$\times \exp\left[-\frac{1}{2v_{T\sigma}^2}\left(v_x^2 + v_y^2 + (v_z - V_\sigma)^2\right)\right], \tag{2}$$

where $B_0$ is the asymptotic value of the magnetic field, $\lambda$ is the sheath thickness, $n_0$ is the sheath peak density, and $v_{T\sigma} = \sqrt{k_B T_\sigma / m_\sigma}$ is the species thermal velocity where $T_\sigma$ and $m_\sigma$ are, respectively, the species temperature and mass. $V_\sigma$ is the species' mean velocity in the $z$ direction, i.e., its drift velocity. It is also assumed that $T_i = T_e := T$ and $V_i = -V_e := V$; the latter can always be made true by choosing a frame of reference where the electrostatic potential $\phi = 0$. Equations (1) and (2) by themselves do not describe an equilibrium. In fact, two conditions must be true in order for this system to be an exact solution of the stationary Vlasov equation: (i) $B_0 = 2\sqrt{\mu_0 n_0 k_B T}$, which describes the balance between the peak magnetic pressure $B_0^2/2\mu_0$ and the peak thermal pressure $n_0 k_B (T_i + T_e) = 2n_0 k_B T$, and (ii) $\lambda = \lambda_D c/V$ where $\lambda_D = \sqrt{\epsilon_0 k_B T / n_0 e^2}$ is the Debye length and $c$ is speed of light, which determines the equilibrium sheath thickness.

The vector potential is chosen to be $\mathbf{A} = -\hat{z}\lambda B_0 \ln \cosh x/\lambda$. Normalizing length by $\lambda$, mass by the species mass $m_\sigma$, and time by $\omega_{c\sigma} = q_\sigma B_0/m_\sigma$ where $q_\sigma$ is the species charge, then a particle obeys Lagrangian dynamics with the normalized Lagrangian $\bar{L} = (\bar{v}_x^2 + \bar{v}_y^2 + \bar{v}_z^2)/2 - \bar{v}_z \ln \cosh \bar{x}$, where barred quantities are normalized to their respective reference units, i.e., $\bar{L} = L/m_\sigma \lambda^2 \omega_{c\sigma}^2$, $\bar{v}_x = v_x/\lambda\omega_{c\sigma}$, and $\bar{x} = x/\lambda$. Because $y$ and $z$ are ignorable coordinates, there are three constants of motion, namely the canonical momenta $\bar{p}_y = \partial \bar{L}/\partial \bar{v}_y = \bar{v}_y$ and $\bar{p}_z = \partial \bar{L}/\partial \bar{v}_z = \bar{v}_z - \ln \cosh \bar{x}$, and the total energy of the particle (recall that $\phi = 0$), $\bar{H} = (\bar{v}_x^2 + \bar{v}_y^2 + \bar{v}_z^2)/2 = \bar{v}_x^2/2 + [\bar{p}_y^2 + (\bar{p}_z + \ln \cosh \bar{x})^2]/2$. The normalized effective potential $\chi(\bar{x})$ of the motion in the $x$ direction is therefore $\chi(\bar{x}) = [\bar{p}_y^2 + (\bar{p}_z + \ln \cosh \bar{x})^2]/2$.

Analyzing the extrema of $\chi(\bar{x})$ shows that it exhibits two shapes depending on the sign of $\bar{p}_z$: (i) a single-well if $\bar{p}_z > 0$ (e.g., black line in Fig. 1d), and (ii) a double-well with a local hill at $\bar{x} = 0$ if $\bar{p}_z < 0$ (e.g., black line in Fig. 1a). In case (ii), if a particle does not have enough energy to overcome the local hill, i.e., $\bar{H} < \chi(0)$ or equivalently $\sqrt{\bar{v}_y^2 + \bar{v}_z^2} = \bar{v}_\perp < -\bar{p}_z$, it oscillates within one of the two wells and does not cross $\bar{x} = 0$. In the opposite case where $\bar{v}_\perp > -\bar{p}_z$, the particle has enough energy to overcome the hill and thus undergoes a full double-well orbit while crossing $\bar{x} = 0$. This double-well orbit class can be further divided into two subclasses depending on the particle's bounce-period-averaged velocity in the $z$ direction $\langle \bar{v}_z \rangle$. Because $\langle \bar{v}_z \rangle = \langle \bar{p}_z \rangle + \langle \ln \cosh \bar{x} \rangle$ while $\bar{p}_z < 0$ is a constant in the case of a double-well $\chi$, a particle can have either a positive or negative $\langle \bar{v}_z \rangle$ depending on its oscillation amplitude in the $x$ direction; particles with higher energies have higher values of $\langle \ln \cosh \bar{x} \rangle$ and thus can have positive values of $\langle \bar{v}_z \rangle$.

Figure 1 summarizes the four classes of particle orbits. The black lines in Fig. 1a–d show the effective potential $\chi$ of each class,

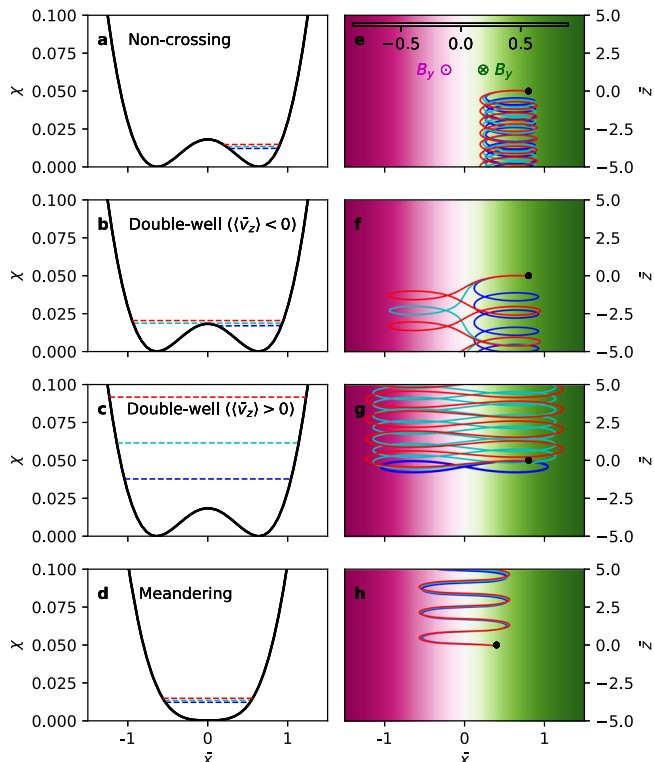

**Fig. 1 Four classes of particle orbits and their effective potentials.**
Effective potentials $\chi$ of the **a** non-crossing (NC) orbit class, **b** double-well orbit class with a negative time-averaged velocity ($\langle \bar{v}_z \rangle < 0$; DW−), **c** double-well orbit class with a positive time-averaged velocity ($\langle \bar{v}_z \rangle > 0$; DW+), and **d** meandering (M) orbit class. **e–h** Particle orbits in the $\bar{x} - \bar{z}$ plane respectively belong to the four classes in **a–d**. The strength of the out-of-plane magnetic field $B_y$ is represented by the magenta and green colors. Three particles are plotted for each class and are labeled by the blue, cyan, and red colors. Each particle's energy is represented by its corresponding color in **a–d**. The blue particles in **f** and **g**, respectively, belong to NC and DW− but are plotted to show the NC→DW− and DW−→DW+ transitions.

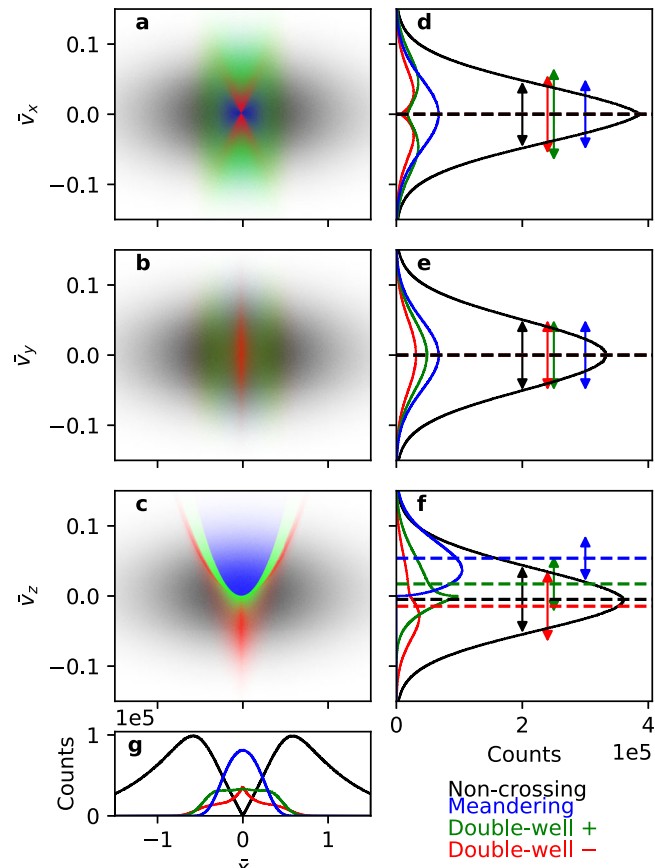

**Fig. 2 Particle distribution in phase space, velocity space, and physical space.** Phase-space distributions of the four orbit classes distinguished by the blue, green, red, and black colors in **a** $\bar{x} - \bar{v}_x$ space, **b** $\bar{x} - \bar{v}_y$ space, and **c** $\bar{x} - \bar{v}_z$ space. Particle histograms in **d** $\bar{v}_x$, **e** $\bar{v}_y$, and **f** $\bar{v}_z$. The dotted lines and arrows are, respectively, the average velocities and two standard deviations of each distribution. **g** Particle histogram in $\bar{x}$.

and the three dashed lines in each panel represent the energies of three particles with differing values of initial $\bar{v}_x$ and thus of $\bar{H}$. Each particle is distinguished by its respective color (blue, red, or cyan). The three lines in Fig. 1e–h show the motion of the three particles in the left panels in the $x-z$ plane, and the black dots represent their starting positions.

Figure 1e represents the non-crossing orbit class[38], hereafter denoted NC, where the particles are simply $\nabla B$ drifting with $\langle \bar{v}_z \rangle < 0$. Figure 1f represents the class where particles undergo full double-well motion with $\langle \bar{v}_z \rangle < 0$, hereafter denoted DW−. The blue particle in Fig. 1f belongs to NC but is plotted to show the transition from NC to DW−. Figure 1g represents the other class where $\langle \bar{v}_z \rangle > 0$, hereafter denoted DW+. Again, the blue particle belongs to the DW− class but is plotted to show the transition from DW− to DW+. Figure 1h represents the meandering or Speiser orbit class[39] with $\langle \bar{v}_z \rangle > 0$, hereafter denoted M.

The DW+ class was previously identified in a context with curved magnetic fields as "cucumber orbits[40,41]" owing to its cucumber shape. Here, we have re-identified the class to clarify the physical origin of such motion and to distinguish more clearly between DW+ and DW−, the latter of which does not exhibit cucumber shapes.

**Phase-space distributions**. Now, let us examine how each orbit class is represented in phase space. $10^8$ particles were randomly

sampled from Eq. (2) with $\bar{V}_\sigma = 0.005$ and $\bar{v}_{T\sigma} = 0.05$—these specific values satisfy the equilibrium condition for the Harris sheet. Figure 2 shows the phase-space distributions (a–c) and velocity space histograms (d–f) in each velocity direction, and Fig. 2g shows the spatial histograms. The orbit classes are distinguished by the black, red, green, and blue colors. The dotted lines and the arrows in the right panels correspond to the mean velocity and the velocity spread (two standard deviations) of each orbit class.

The phase-space distribution of each orbit class has its own contribution to current sheet density, temperature, and strength. The spatial distribution in Fig. 2g is related to the density, and the spreads and means of the velocity distributions in Fig. 2d–f are, respectively, related to the temperature and current strength of each orbit class.

Figure 2d shows that the velocity spread and hence the temperature in the $x$ direction, $T_{xx}$, has the following hierarchy: NC<DW−<DW+. This is because the transition from NC to DW necessarily involves a passage through the unstable equilibrium as in Fig. 1b, which in turn involves a breakdown of adiabatic invariance and phase-mixing[42]. $T_{xx}$ of the M class is equal to the overall equilibrium temperature. The mean velocity in the $x$ direction is befittingly zero for all classes owing to symmetry.

Figure 2e shows that all classes have the same temperatures and zero mean velocities in the $y$ direction, since $\bar{v}_y$ is a constant of motion.

Figure 2f shows that the temperature in the $z$ direction has the hierarchy M<DW+<NC<DW−. The NC and DW− classes have

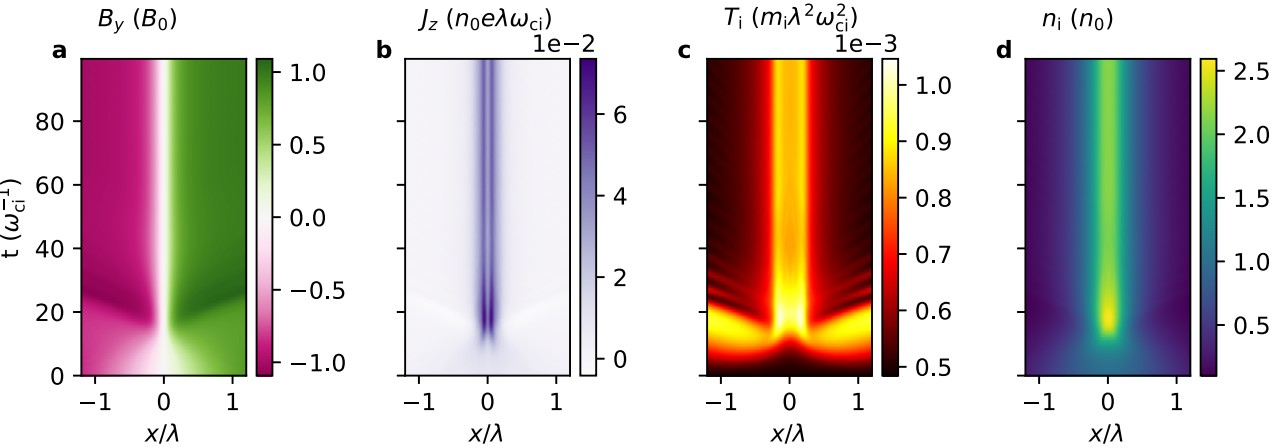

**Fig. 3 Streak plots of variables from the particle-in-cell simulation.** Streak plots of **a** the sheared magnetic field $B_y$, **b** the current density $J_z$, **c** the ion temperature $T_i$, and **d** the ion density $n_i$ from $t = 0$ to $100\omega_{ci}^{-1}$.

negative mean velocities and the other two have positive mean velocities.

Figure 2g shows that the three non-NC classes are spatially concentrated near the center. Also, the DW classes have relatively flat-top density profiles compared with the M class, a trait which will be revisited later.

**Equilibration process**. We now have all the ingredients to understand how an initially disequilibrated current sheet equilibrates. Let us consider an under-heated Harris current sheet with a temperature lower than its equilibrium value. In this case, because the thermal pressure at the center is lower than the magnetic pressure at the outskirts, one expects heating and pinching (increase of current density) of the current sheet that lead to an equilibrium.

Let us first predict how the heating and pinching will happen. Inserting Eqs. (1) and (2) in the time-dependent Vlasov equation yields

$$\frac{\partial \ln f_\sigma}{\partial \bar{t}} = -2\left(\frac{\bar{V}_\sigma}{2\bar{v}_{T\sigma}^2} - 1\right)\bar{v}_x \tanh \bar{x}. \tag{3}$$

The two aforementioned conditions for equilibrium $-B_0 = 2\sqrt{\mu_0 n_0 k_B T}$ and $\lambda = \lambda_D c/V$—together reduce to $V_\sigma \lambda \omega_{c\sigma} = 2v_{T\sigma}^2$ or equivalently $\bar{V}_\sigma = 2\bar{v}_{T\sigma}^2$, for which Eq. (3) yields the stationary solution $\partial f_\sigma/\partial \bar{t} = 0$. However, if the sheet is under-heated so that $2\bar{v}_{T\sigma}^2 < \bar{V}_\sigma$, then the quantity $\xi := \bar{V}_\sigma - 2\bar{v}_{T\sigma}^2$ is positive and Eq. (3) yields a solution linear in a small time interval $\Delta \bar{t}$:

$$f_\sigma \propto \exp\left[-\frac{1}{2\bar{v}_{T\sigma}^2}\left(\bar{v}_x + \xi \Delta \bar{t} \tanh \bar{x}\right)^2\right]. \tag{4}$$

The mean velocity in the $x$ direction is thus $\bar{V}_x(\bar{x}) = -\xi \Delta \bar{t} \tanh \bar{x}$. At positive $\bar{x}$, particles gain negative $\bar{v}_x$ and vice-versa; therefore, the initial linear response of an under-heated Harris sheet is to bring particles closer to the center by increasing their $|\bar{v}_x|$.

This response induces transitions among particle classes. It is apparent from Fig. 1a–c that an increase in $|\bar{v}_x|$ moves NC particles to DW− and DW− particles to DW+. Applying the analysis of the phase-space distributions of the four classes in Fig. 2, these class transitions explain (i) current sheet heating in the $x$ direction, and (ii) current sheet pinching due to increases in both density and mean velocity at the center (note that the velocity decrease from the NC→DW− transition is more than compensated for in the DW−→DW+ transition). Also, there is

no transition to or from the M class because the shape of $\chi$ is such that a change in $\bar{v}_x$ does not induce orbit class transitions.

The above analysis only considers linear dynamics assuming that the current sheet profile remains stationary. It is therefore not valid in the nonlinear regime where the profile self-consistently changes along with orbit class transitions. However, we may infer from the analysis the primary mechanism—at least in the linear regime—underlying current sheet heating and pinching: transitions from the NC class to the DW classes and no transitions to or from the M class.

These predictions will now be verified with a one-dimensional particle-in-cell simulation. The initial condition was an under-heated Harris current sheet with a temperature $T = 0.2T_{eq}$ where $T_{eq} = B_0^2/(4\mu_0 n_0 k_B)$ is the Harris equilibrium temperature. The initial sheet thickness was $\lambda = 10d_i$ where $d_i$ is the collisionless ion skin depth. Figure 3 shows streak plots of $B_y$, the current density $J_z$, the ion temperature $T_i$, and the ion density $n_i$. The current sheet pinches and heats up until $\sim 30\omega_{ci}^{-1}$, after which it remains steady and thus reaches equilibrium.

Figure 4a–c show $f_i$ in $x–v_x$ space at $t = 0, 10, 100\omega_{ci}^{-1}$, respectively. Figure 4b confirms the initial response of the under-heated current sheet as predicted by Eq. (4), namely the focussing of the particles towards the center. Figure 4c shows the equilibrium reached by $f_i$, and Fig. 4d shows the difference ($\Delta f_i$) between the initial state (Fig. 4a) and the equilibrium state (Fig. 4c). Comparing Fig. 4d to Fig. 2a, it is apparent that the NC class de-populates and migrates to the DW classes. The dynamics in the simulation is fully nonlinear, so transition to the M class also occurs, albeit less significantly than the main NC→DW transition.

Figure 4e–g, i are the same as Fig. 4a–d except that they are in $x–v_z$ space. Again, the NC→DW transition is evident from a comparison to the pronounced Y-shape of the phase-space distribution of the DW classes (Fig. 2c). Therefore, we have confirmed that collisionless equilibration of an under-heated Harris current sheet is mainly due to orbit class transitions from NC to DW.

It is clear from Fig. 4c, g that the final equilibrium is most naturally described by the relative population in each orbit class, rather than, e.g., Maxwellian or kappa distributions. Figure 4h shows the distribution in $v_z$ at $x = 0.2\lambda$, whose profile is clearly non-Maxwellian.

Note that electrons also mainly transition from NC to DW in this process because the orbit classes apply generally for any species. Figure 4 is therefore qualitatively applicable to electrons, except that their velocities change signs owing to their negative charge.

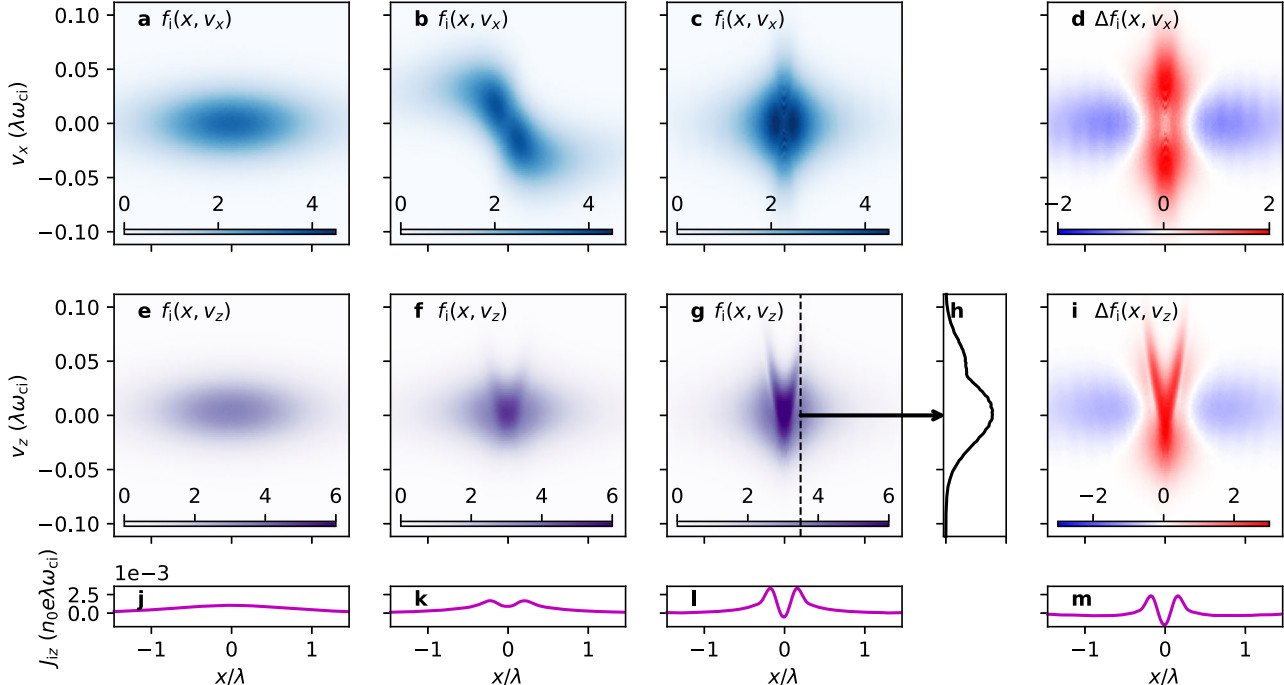

**Fig. 4 Time evolution of the ion distribution function from the particle-in-cell simulation.** Ion distribution function $f_i$ in $x-v_x$ space at **a** $t = 0$, **b** $t = 10\omega_{ci}^{-1}$, and **c** $t = 100\omega_{ci}^{-1}$. **d** The difference ($\Delta f_i$) between $f_i$ in **c** and **a**. **e–g** and **i** are respectively the same as **a–c** and **d**, except in $x-v_z$ space. **h** A slice through the dotted line in **g**. **j–l** The ion current density $J_{iz}$ obtained by taking the first velocity moment of **e–g**. **m** The difference between **j** and **l**.

**Origin of bifurcated current sheets.** Figure 4j–l show the time evolution of the ion current density in the $z$ direction, $J_{iz}$, and Fig. 4m shows the difference between the initial and final $J_{iz}$. The bifurcated structure is evident, which naturally arises from the pronounced Y-shape of the phase-space distribution of the DW classes to which particles migrate from the NC class. The total current density—mainly carried by the electrons—is also bifurcated, as shown in Fig. 3b. Bifurcated current structures are thus natural by-products of the collisionless equilibration process of a current sheet.

Let us compare the simulation results with a current sheet detected by the MMS mission[43] from 20:24:00 to 20:24:15 UT on 17 June 2017, when the spacecraft was located at $(-19.4, -10.4, 5.5)R_E$—where $R_E$ is the Earth's radius—in Geocentric Solar Ecliptic (GSE) coordinates while crossing the magnetotail plasma sheet from the southern to the northern hemisphere. This sheet has also been examined in previous studies under different contexts[37,44,45] and has a half-width of ~67 km, which is ~$9d_e$, where $d_e = 7.5$km is the electron skin depth based on the background electron density $n_e = 0.5$cm$^{-3}$[37]. Because the half-width of the observed sheet is much less than the ion skin depth $d_i$, its dynamics is mainly controlled by electrons.

The half-width of the simulated sheet pinches down to ~$0.1\lambda$ $= 1d_i = 10d_e$ (because $m_i/m_e = 100$ was used in the simulation), as can be seen in Fig. 3a. The simulated sheet and the observed sheet are thus similar in that their widths are ~$10d_e$ and $\lesssim 1d_i$, so their sheet dynamics are mainly controlled by electrons. Therefore, the focus will now be on electrons instead of ions because (i) both sheets have electron-scale thicknesses, and (ii) as a confirmation that electrons have similar orbit class transition dynamics to that of ions.

Figure 5 shows a side-by-side comparison of the current sheet detected by MMS and that from the particle-in-cell simulations. The data are presented using a local coordinate system, $LMN$. The sheared magnetic field is in the $L$-direction, and $M$ and $N$ are, respectively, parallel and normal to the current sheet. The current

is carried mainly by the electrons in both the observation and the simulation. The finite electron outflow $v_{eL}$ in Fig. 5b indicates that the observed current sheet is undergoing magnetic reconnection, whereas the simulated current sheet, being one-dimensional, is not. Reconnection induces perpendicular electron heating at the sheet center[46], which explains the central increase of $T_{eMM}$ in Fig. 5d relative to Fig. 5j. The relative increase of $T_{eLL}$ at the outskirts in Fig. 5d is also attributed to reconnection-induced parallel electron heating[46].

Apart from such reconnection-related dissimilarities, the observed and simulated profiles agree strikingly well, including the bifurcated current structure. In particular, the simulated equilibrium explains the central dip and increased outskirts of the electron temperature tensor element $T_{eMM}$ relative to $T_{eNN}$, as shown in Fig. 5d, j. The profile of $T_{eNN} - T_{eMM}$ in Fig. 5e is remarkably reproduced by Fig. 5k, except for the relative central dip in Fig. 5e owing to the reconnection-induced increase of $T_{eMM}$. Same goes for the pressure tensor elements $P_{eMM}$, $P_{eNN}$, and $P_{eNN} - P_{eMM}$ (Fig. 5f, l).

The increased amount of electron population in the DW classes is shown not only by the current, temperature, and pressure structures, but also by the density plateau in Fig. 5c, i which is due to migrations to the DW classes (see Fig. 2g). This density plateau was also observed in Cluster measurements[29].

## Discussion

Although the kinetic equilibrium attained by relaxation has been presented as an example of bifurcated current sheets, we are not claiming that all such sheets are in equilibrium states. Instead, the claim is that bifurcated structures are natural repercussions of the collisionless current sheet equilibration process and so are likely to be observed in a variety of phenomena as the underlying structure. As mentioned in the Introduction, numerous explanations for bifurcated current sheets have been put forth; these explanations will now be unraveled in relation to the relaxation process.

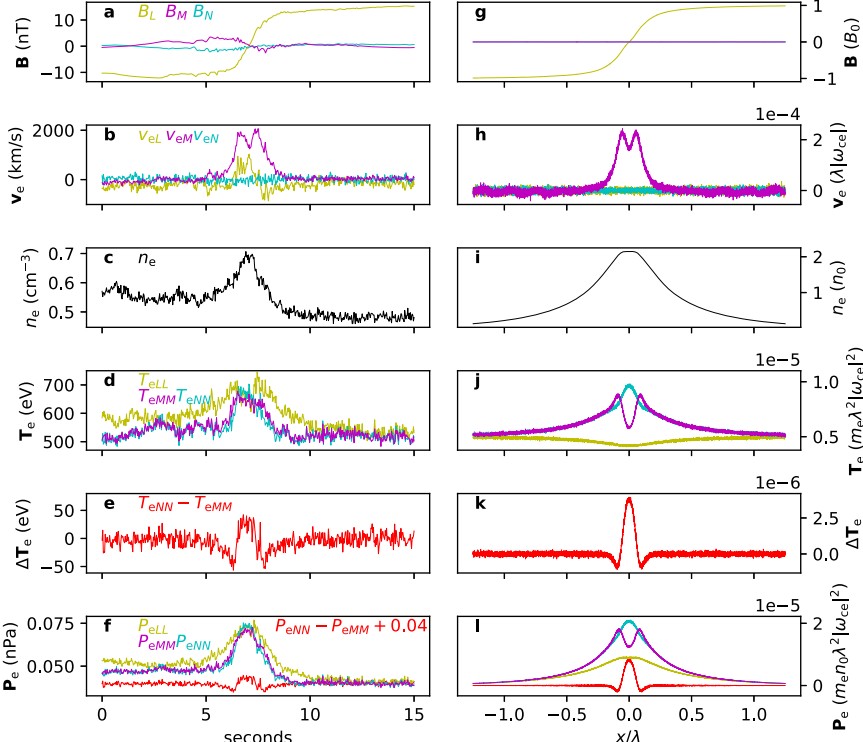

**Fig. 5 Comparison of a current sheet detected by MMS to that from the particle-in-cell simulation. a–f** Sequentially, the magnetic field **B**, electron velocity $\mathbf{v}_e$, electron density $n_e$, diagonal elements of the electron temperature tensor $\mathbf{T}_e$, the difference between the temperature tensor elements $T_{eNN}-T_{eMM}$, and diagonal elements of the electron pressure tensor $\mathbf{P}_e$ and $P_{eNN}-P_{eMM}$ (shifted up by 0.04) detected by the Magnetospheric Multiscale spacecraft from 20:24:00 to 20:24:15 UT on 17 June 2017. The x axis is seconds from 20:24:00 UT. The current sheet crossing velocity is ~67 km/s, so one second corresponds to 67 km, or $9d_e$. **g–l** Quantities from the particle-in-cell simulation respectively corresponding to **a–f**. The x axis is in units of $\lambda = 10d_i = 100d_e$.

Magnetic reconnection has been one of the proposed causes of bifurcated current sheets[30,47], but many such sheets were also observed without any fast flows[29,48], which are signatures of reconnection. Nevertheless, a statistical study indicated that the thinner the structures are, the more likely they are to be bifurcated[28]. These observations can be unified by the fact that thinner current sheets are more likely to involve sub-skin-depth collisionless dynamics, which is favorable for the occurrence of both collisionless reconnection and the present collisionless relaxation process. A possible scenario is one where an initially thick, under-heated current sheet equilibrates to a thin, sub-skin-depth bifurcated structure, which then undergoes collisionless reconnection. In fact, the initial condition for reconnection in collisionless situations is more likely to be the equilibrium presented here than widely used specific solutions such as the Harris sheet[49]. If the sheet is not thin enough for reconnection to occur, then it may remain bifurcated and steady.

Flapping motion was also observed in conjunction with bifurcated current sheets[29,31]. This motion involves fast thinning and thickening of the sheet[31]. Such fast motion will naturally induce bifurcation via two possible scenarios: (i) disequilibration of current sheets, followed by relaxation via spontaneous orbit classes transitions, or (ii) unspontaneous transitions driven by the external source that thins the sheet.

Equilibria involving anisotropic temperatures have also been shown to exhibit bifurcated structures[17,32,33], but the source of the anisotropy was not clear so the amount of anisotropy had been set ad hoc. The present collisionless relaxation process naturally induces temperature anisotropy, which is thus an innate result of the equilibration process rather than a cause of bifurcated structures.

Speiser motion (M class) was also attributed to bifurcated structures[21,34]. However, it is clear from Fig. 2 that the M class cannot contribute to bifurcated structures if the density is peaked near the center, unless the density itself is bifurcated[21] and/or heavier species are taken into account[34].

Some studies[20,40,41] invoke non-adiabatic scattering of particles from M to DW+ via a slow diffusive process in current sheet equilibria. However, the diffusion coefficient of such process is zero for $B_x = 0$ because it scales with the curvature parameter $\kappa$ (cf. equation 7 of Zelenyi et al.[41]). We have shown here that neither curved magnetic fields nor diffusive processes are necessary; simply choosing a disequilibrated initial state is sufficient for the development of bifurcated structures, although diffusion due to field curvature may aid the process.

In summary, the collisionless relaxation process of a disequilibrated current sheet was studied. The process is most naturally understood by orbit class transitions, which were analytically predicted and numerically verified. The relaxation mechanism was identified as the origin of bifurcated current sheets, and the significance of this identification in regards to previous explanations of bifurcated structures was discussed.

## Methods

**Sampling from and categorization of the Harris distribution function**. Particle positions and velocities were sampled from Eq. (2) using the `numpy.random` package in Python 3.8. Particles with $\bar{p}_z > 0$ and $\bar{p}_z < -\bar{v}_\perp$ were categorized into M and NC, respectively. For the rest of the particles that belong to the DW classes, the following steps were taken to further categorize them.

First, a simple analysis of the Hamiltonian of each particle shows that its oscillation amplitude in the x direction is given by $\bar{x}_{\max} = \operatorname{arccosh}\left(\exp\left[\bar{v}_\perp - \bar{p}_z\right]\right)$.

The bounce-period-averaged $\bar{v}_z$ of the particle is then given by

$$\langle \bar{v}_z \rangle = \frac{2}{T_0} \int_{-\bar{x}_{max}}^{\bar{x}_{max}} \frac{\bar{v}_z}{\bar{v}_x} d\bar{x}, \tag{5}$$

$$= \frac{2}{T_0} \int_{-\bar{x}_{max}}^{\bar{x}_{max}} \frac{\bar{p}_z + \ln \cosh \bar{x}}{\sqrt{\left(\bar{p}_z + \ln \cosh \bar{x}_{max}\right)^2 - \left(\bar{p}_z + \ln \cosh \bar{x}\right)^2}} d\bar{x}, \tag{6}$$

where $T_0 = 2 \int_{-\bar{x}_{max}}^{\bar{x}_{max}} d\bar{x}/\bar{v}_x$ is the bounce period. Only the sign of $\langle \bar{v}_z \rangle$ matters here, so the integral in Eq. (6) was evaluated numerically for each particle using the scipy.integrate.quad package in Python 3.8. Particles with positive $\langle \bar{v}_z \rangle$ were categorized into DW+, and the rest into DW−.

**Particle-in-cell simulation.** The open-source, fully-relativistic particle-in-cell code, SMILEI[50], was used. The 1D simulation domain was $10\lambda = 100d_i$ long and was divided into $2^{15} = 32,768$ grid points. Open boundary conditions (Silver–Müller) were employed for the electromagnetic fields, and periodic boundary conditions were imposed for the particles. In all, 10,000 particles were placed per cell per species, so ~$6 \times 10^8$ particles were simulated with a mass ratio $m_i/m_e = 100$. The simulation run with a frequency ratio of $\omega_{ce}/\omega_{pe} = 5$ is shown in this paper for clarity of presentation; ratios as low as $\omega_{ce}/\omega_{pe} = 0.2$ were also tried, but lower ratios simply increased the duration of plasma oscillations that either damp or travel away from the center without any noticeable effect on the core relaxation mechanism. The initial conditions were Eqs. (1) and (2), and the electrostatic potential $\phi = 0$. The initial temperature was set as one-fifth of the Harris equilibrium temperature, i.e., $T = 0.2 T_{eq}$ where $T_{eq} = B_0^2/(4\mu_0 n_0 k_B)$ is the temperature that yields the Harris equilibrium. The simulation time was $t_{max} = 100\omega_{ci}^{-1}$ with a time step $\Delta t = 7.63 \times 10^{-4} \omega_{ci}^{-1}$.

The simulations were run on the KAIROS computer cluster at Korea Institute of Fusion Energy.

**MMS data and local LMN coordinates.** Data from MMS2, MMS3, and MMS4 from 20:24:00 to 20:24:15 UT on June 17, 2017 were averaged to yield the profiles shown in Fig. 5a–e. MMS1 data were omitted because the current density did not exhibit an obvious bifurcated structure. The magnetic field data were collected by the Fluxgate Magnetometer instrument[51] and the plasma data by the Fast Plasma Investigation instrument[52]. The local LMN coordinate system is obtained from a minimum variance analysis[53] of the averaged magnetic field data which are in GSE coordinates. The values of the unit vectors are $L = (0.942, 0.308, -0.130)$, $M = (0.194, -0.189, 0.963)$, and $N = (0.272, -0.932, -0.238)$ in GSE coordinates. $L$ is the direction of the sheared magnetic field, $N$ is the direction normal to the current sheet, and $LMN$ form a right-handed coordinate system.

## Data availability

MMS data are publicly available from https://lasp.colorado.edu/mms/sdc/public. The data from the PIC simulations are available from https://doi.org/10.5281/zenodo.4607112.

## Code availability

SMILEI[50] is an open-source particle-in-cell code available from https://smileipic.github.io/Smilei. MMS data were analyzed using the pySPEDAS package[54], available from https://github.com/spedas/pyspedas. The codes used in the data analyses are available from Y.D.Y. upon reasonable request.

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

## Acknowledgements

This work was supported by the National Research Foundation of Korea under grant nos. NRF-2019R1C1C1003412, NRF-2019R1A2C1004862, and 2019M1A7A1A03088456. We thank the entire MMS team and MMS Science Data Center for providing the high-quality data for this study. We also thank J.M. Kwon, J.H. Kim, and Korea Institute of Fusion Energy for providing the computer resources.

## Author contributions

Y.D.Y. conceived the central idea, performed the simulations and theoretical analysis, analyzed the spacecraft data, and wrote the manuscript based on extensive discussions with G.S.Y. D.E.W. contributed to the interpretation of the simulation and observation results, as well as to the revision of the draft. J.L.B. oversaw the MMS project and provided general guidance.

## Competing interests

The authors declare no competing interests.
