## [Peer Review File · Nature Communications]

Reviewers' Comments:

Reviewer #1:

Remarks to the Author:

This paper presents a reasonable explanation for the origin of bifurcated current sheets. The division of particle orbits into separate orbit classes and the transition across orbit classes to another is hypothesized to cause the bifurcated sheets, in the process of equilibrating. As such it is a worthy contribution to the field.

My only observation is that the authors claim to insert equations 1 and 2 obtained from an equilibrium solution to the Vlasov equation, into the Vlasov equation, and obtain equation 3, a time dependent relaxation equation, therefore non equilibrium. How can this be? They need to be clear here.

The paper should be returned to the authors for re-submission after revision.

Edmund Spencer

Reviewer #2:

Remarks to the Author:

The authors investigate the collisionless relaxation process of a disequilibrated current sheet and demonstrated analytically and numerically that it is most naturally understood by orbit class transitions. The arguments the authors give and the numerical simulations presented are convincing and novel but, it is not a priori evident for a non-expert why the subject is important to others in the space and/or plasma physics community. I would recommend that the authors address this question with more care before publication so the results can be appealing to a broad audience.

Reviewer #3:

Remarks to the Author:

This study is about the relaxation of plasma current sheets and the formation of bifurcated current sheets. The authors make use of analytical solutions of orbital classes of particles near a current sheet and find, with the help of a particle-in-cell plasma simulation, that the bifurcated structure of a relaxing current sheet arises due to transitions between the orbital classes. The authors finally compare their simulation results to a bifurcated current sheet observed by the MMS spacecraft.

The topic of the formation bifurcated current sheets an important and long-standing question in space and plasma physics. The combined approach of analytical work, simulation and observations provide strong evidence for the suggested formation of the bifurcated structure. The analysis in the paper is thorough and the presentation is clear. I only have a small number of minor comments and suggestions.

1. Regarding the comparison between the simulation and the MMS observations. The authors mention that the current sheet observed by MMS is significantly thinner than in the simulation. Can the authors expand on this difference? Is the current sheet close to equilibrium? Does reconnection contribute to thinning the current sheet?

2. On the same note, the authors should clarify how the x-axes on the two sides in Figure 5 compare. Since the speed and thickness of this current sheet are known from before (i.e., Wang et al., 2020), it should be possible to provide a conversion factor between seconds and " x/λ " in the observations.

3. It would be more illustrative if the color scale in Figure 4d & 4i was centered around zero (so that 0 is white).
4. Line 195: It seems the variable "Re" for Earth radius is not explained.
5. Line 199: It seems the variable "de" for electron skin depth is not explained.
6. Line 238: "It fact" -> "In fact"
7. Line 303: The authors should clarify that it is the magnetic field data which are used to construct the LMN coordinate system. Also, I don't think it's prudent to call the calibrated data in GSE "raw".

Response to Reviewer Comments

We tremendously thank all three reviewers for their careful reading of our manuscript and their generally positive opinion of it. The manuscript has been revised according to the reviewers' comments (changes marked in RED), which in our opinion have greatly improved the manuscript. Please see below our detailed point-by-point responses.

REVIEWER COMMENTS

Reviewer #1 (Remarks to the Author):

This paper presents a reasonable explanation for the origin of bifurcated current sheets. The division of particle orbits into separate orbit classes and the transition across orbit classes to another is hypothesized to cause the bifurcated sheets, in the process of equilibrating. As such it is a worthy contribution to the field.

My only observation is that the authors claim to insert equations 1 and 2 obtained from an equilibrium solution to the Vlasov equation, into the Vlasov equation, and obtain equation 3, a time dependent relaxation equation, therefore non equilibrium. How can this be? They need to be clear here.

The paper should be returned to the authors for re-submission after revision.

Edmund Spencer

Thank you for pointing out an important source of possible confusion. Equations 1 and 2 by themselves do not describe an equilibrium; the two conditions mentioned shortly after Eq. 2 must be satisfied for the system to be in equilibrium. The simultaneous satisfaction of the two conditions leads to the relation $\bar{V}_\sigma = 2\bar{v}_{T\sigma}^2$, for which Eq. 3 describes an equilibrium. If $\bar{V}_\sigma \neq 2\bar{v}_{T\sigma}^2$, then Eqs. 1 and 2 do not describe an equilibrium.

The above discussion has been reflected in the manuscript in lines 68-69 and 146-149.

Reviewer #2 (Remarks to the Author):

The authors investigate the collisionless relaxation process of a disequilibrated current sheet and demonstrated analytically and numerically that it is most naturally understood by orbit class transitions. The arguments the authors give and the numerical simulations presented are convincing and novel but, it is not a priori evident for a non-expert why the subject is important to others in the space and/or plasma physics community. I would recommend that the authors address this question with more care before publication so the results can be appealing to a broad audience.

We thank the reviewer for his/her positive and constructive comments. We have expanded the Introduction to explain further why the subject matter is puzzling, interesting, and important.

In the first paragraph, we explain further the role current sheets play, as well as list examples of important dynamic events to which current sheets act as underlying structures.

In the second paragraph, we explain further why a plasma system in general should undergo equilibration processes, and why the knowledge of the process is important.

In the third paragraph, we explain why the recurrent observations of bifurcated current sheets are surprising by contrasting it with a single-peaked current sheet, which was thought to be the norm. This implies that something is missing from the original picture and thus further efforts are warranted.

Reviewer #3 (Remarks to the Author):

This study is about the relaxation of plasma current sheets and the formation of bifurcated current sheets. The authors make use of analytical solutions of orbital classes of particles near a current sheet and find, with the help of a particle-in-cell plasma simulation, that the bifurcated structure of a relaxing current sheet arises due to transitions between the orbital classes. The authors finally compare their simulation results to a bifurcated current sheet observed by the MMS spacecraft.

The topic of the formation bifurcated current sheets an important and long-standing question in space and plasma physics. The combined approach of analytical work, simulation and observations provide strong evidence for the suggested formation of the bifurcated structure. The analysis in the paper is thorough and the presentation is clear. I only have a small number of minor comments and suggestions.

1. Regarding the comparison between the simulation and the MMS observations. The authors mention that the current sheet observed by MMS is significantly thinner than in the simulation. Can the authors expand on this difference? Is the current sheet close to equilibrium? Does reconnection contribute to thinning the current sheet?

This is an excellent point, and we agree that the comparison between the observed sheet and the simulated sheet should be discussed more. However, we did NOT originally mean to say that the observed sheet is significantly thinner than the stabilized sheet (final state) in the simulation. The observed sheet has a half-width of around $9d_e \ll d_i$ where d_i and d_e are ion and electron skin depths, respectively. The simulated sheet starts from a half-width of $10d_i$, but pinches down to about $1d_i$ or $10d_e$ (a mass ratio of 100 was used in the simulation) where it stabilizes. Therefore, both sheets have the same thickness in terms of d_e , but the simulated sheet is thicker in terms of d_i . Nevertheless, the electron dynamics in both sheets are expected to be similar because both sheets have half-thicknesses of $\leq d_i$.

The above discussion has been reflected in the manuscript in lines 207-215.

In our opinion, it is difficult to tell just from the present observation whether the current sheet is close to an equilibrium, or whether reconnection contributes to thinning the current sheet, without extensive additional work. Two possibilities come to mind: (i) the current sheet has been forced by reconnection to thin and has achieved a new pseudo-equilibrium, or (ii) the current sheet was in such equilibrium state as the simulated sheet, but a perturbation has induced reconnection. The question of which case the observed current sheet corresponds to is an open one and is possibly subject to future investigations.

2. On the same note, the authors should clarify how the x-axes on the two sides in Figure 5 compare.

Since the speed and thickness of this current sheet are known from before (i.e., Wang et al., 2020), it should be possible to provide a conversion factor between seconds and " x/λ " in the observations.

This would be an excellent clarification. However, λ in the simulation by itself does not really mean much because it is the initial sheet thickness. Because of equilibration, the sheet pinches to a much thinner width. Therefore, it would be more useful if "seconds" in the observation and " x/λ " in the simulation could be converted into physically meaningful scales such as the electron skin depth.

As such, we have specified this conversion in the caption of Figure 5. This conversion is also easily inferable from the answer to the previous point (lines 207-215).

3. It would be more illustrative if the color scale in Figure 4d & 4i was centered around zero (so that 0 is white).

This is a good point and the change has been made in Figure 4.

4. Line 195: It seems the variable " R_e " for Earth radius is not explained.

5. Line 199: It seems the variable " d_e " for electron skin depth is not explained.

Thank you for spotting the undefinitions. The variables R_e and d_e are now properly explained (line 204, 208)

6. Line 238: "It fact" -> "In fact"

The typo was fixed.

7. Line 303: The authors should clarify that it is the magnetic field data which are used to construct the LMN coordinate system. Also, I don't think it's prudent to call the calibrated data in GSE "raw".

We thank the reviewer for pointing this out and completely agree. The word "raw" was replaced by "magnetic field" so that both points are addressed at the same time (lines 318).

Reviewers' Comments:

Reviewer #1:

Remarks to the Author:

The paper looks good now. My concerns have been addressed. I have read the other reviewers comments and critique, and find the authors replies and modifications to the paper satisfactory.

The paper is ready for publication.

E. Spencer

Reviewer #3:

Remarks to the Author:

The authors have done a great job answering my concerns and have done the appropriate changes. I have no further comments.

Response to Reviewer Comments

Reviewer #1 (Remarks to the Author):

The paper looks good now. My concerns have been addressed. I have read the other reviewers comments and critique, and find the authors replies and modifications to the paper satisfactory.

The paper is ready for publication.

E. Spencer

Reviewer #3 (Remarks to the Author):

The authors have done a great job answering my concerns and have done the appropriate changes. I have no further comments.

We tremendously thank the reviewers for their decision to accept the manuscript. Their previous comments were very constructive and have resulted in a greatly improved manuscript.

Young Dae Yoon